

# Measurement report: Spectral actinometry at SMEAR-Estonia

Andres Kuusk[1] and Joel Kuusk[1]

[1]Tartu Observatory, University of Tartu, 61602 Tõravere, Estonia

**Correspondence:** Andres Kuusk (Andres.Kuusk@to.ee)



**Abstract.**

Systematic spectral measurements of downwelling solar radiation, both of global and diffuse, have been collected in summertime during 8 years in the hemi-boreal zone in south east Estonia near the SMEAR-Estonia research station. The measurements provided information about the variation of spectral and total fluxes of downwelling hemispherical global and diffuse solar radiation in the wavelength range from 300 to 2160 nm with spectral resolution of 3 nm in UV to 16 nm in SWIR spectral regions. Unique data have been collected and quantitative description of the variability of the measured spectra is provided. For the description of the synoptic situation during measurements, instead of cloud cover, the ratio of measured to possible total flux in the spectral range of 320–1800 nm is used. This ratio could be used as the primary meteorological parameter instead of cloud cover which is difficult to measure instrumentally.

## 1    Introduction

Measurements of global and diffuse downwelling radiation are carried out in Estonia at Tartu-Tõravere actinometric station of Estonian Weather Service since 1955 (Russak and Kallis, 2003). Both, global and diffuse irradiance are measured with pyranometers which are spectrally integrating sensors. While in the studies of energy fluxes in the atmosphere and at Earth's surface global fluxes are needed, in ecological studies of the vegetation-atmosphere interaction the energy distribution over the radiation spectrum is important. John et al. (2013) studied the allometry of cells and tissues within leaves, and found that future work is needed to consider the possible influence of the environment. Many trait allometries shift substantially due to plasticity across different growing conditions, i.e., different supplies of light, nutrients, and/or water. The study by Solomakhin and Blanke (2010) showed how the changes in the spectrum of incident light due to colored hailnets affect leaf anatomy, vegetative and reproductive growth as well as fruit coloration in apple. Ji et al. (2020) found that solar radiation components photosynthetically active radiation and ultraviolet-B have different associations with leaf nitrogen and phosphorus content. The direct associations, when solar radiation is indicated by spectral components, are greater than the indirect associations. So when predicting the effects of global dimming on ecosystem nutrient fluxes, the roles of direct, diffuse, and spectral components of solar radiation must be distinguished. The study by Moon et al. (2020) concludes that accurate and spectrally resolved canopy radiative transfer models are critically necessary to realistically determine chemical reactions and gas concentrations within plant canopies and in the immediately overlying atmospheric boundary layer.

The spectral composition of extraterrestrial solar radiation is monitored on-board of satellites (Harder et al., 2005, 2010; Kopp, 2014). For designing photoelectric solar power equipment some episodic measurements of irradiance spectra at Earth's surface have been carried out (Norton et al., 2015), and model simulations with atmospheric radiative transfer models for standard situations are provided (Renewable Resource Data Center (RReDC), 2003; Gueymard, 2004). Eddy covariance (EC) sites for measuring fluxes of trace gases usually include optical sensors measuring downwelling, and sometimes also upwelling shortwave radiation, however, a review by Balzarolo et al. (2011) revealed that only 5 out of 40 European EC sites involved in the study had hyperspectral radiometers while the majority used multispectral or broadband sensors.





In 2013–2015 a Station for Measuring Ecosystem-Atmosphere Relations, SMEAR-Estonia was established in south east Estonia at the Järvselja Experimental Forestry station (SMEAR-Estonia, 2015). The station completes the network of Finnish SMEAR-stations (SMEAR-Estonia, 2015). The coordinates of the SMEAR-Estonia are (58° 16' 6" N, 27° 18' 16" E). For providing the research program of the SMEAR station with radiation data a special spectrometer SkySpec was designed. Measurements of global and diffuse solar radiation with SkySpec started at SMEAR-Estonia in August 2013. Measurements are carried out during vegetation period.

## 2   SkySpec spectrometer

SkySpec spectrometer is a purpose-built instrument for continuous measurement of global and diffuse downwelling spectral irradiance in the shortwave spectral domain covering with three spectrometer modules the wavelength range of 295–2205 nm. Spectral resolution is 3 nm in ultraviolet (UV), 10 nm in visible and near infrared (VNIR), and 16 nm in shortwave infrared (SWIR) spectral region. The spectral range is covered by 541 sensors. The standard scenario used for field measurements includes the following stages: dark signal (1 minute), global irradiance (5 minutes), diffuse irradiance (1 minute), global irradiance (5 minutes). This scenario is repeated in continuous loop when the solar zenith angle $SZA < 90°$. For $90° < SZA < 95°$ diffuse measurement stage is omitted. No measurements are done if $SZA > 95°$. If irradiance levels are very low and integration time of any of the spectrometer modules exceeds the duration of current measurement stage, the stage is extended and at least one measurement is acquired with all three spectrometer modules. Integration time is automatically adjusted independently for each spectrometer module so that the maximal value of the recorded raw signal is between 60% and 90% of the full scale value. Maximal usable integration times are limited by dark signal levels and are set to 200 s, 250 s, and 1.5 s for the UV, VNIR, and SWIR modules of the spectrometer, respectively. Dark measurements are made by covering the fore-optics with a mechanical shutter. A shadow disk is used for blocking approximately 6° in the direction of the Sun during the diffuse measurements. An azimuth-elevation tracking system is used for moving the shadow disk with complete fore-optics tracking the Sun in azimuthal direction. The detailed description of the spectrometer, calibration and metrological processing of data is provided by Kuusk and Kuusk (2018).

## 3   Data

The summary of SkySpec measurements at Järvselja is in Table 1. From 2013 to 2016 the spectrometer was installed on an open place on ground at the distance of 1 km from the SMEAR-Estonia. The cover of horizon at the spectrometer is shown in Fig. 2. Since 2017 the SkySpec is installed near the SMEAR-Estonia atop a 30-m tower at the distance of 45 m from the SMEAR tower. Thus the horizon is open in all directions except the 130 m tower covers low sun for a while in spring and autumn.

For the special analysis the spectra of global radiation in case of clear sky are extracted for the sun zenith angles of 40°, 45°, 50°, 55°, and 60° at 530 wavelengths in the range of 300.1 to 2160.9 nm. The total number of such spectra in every summer is





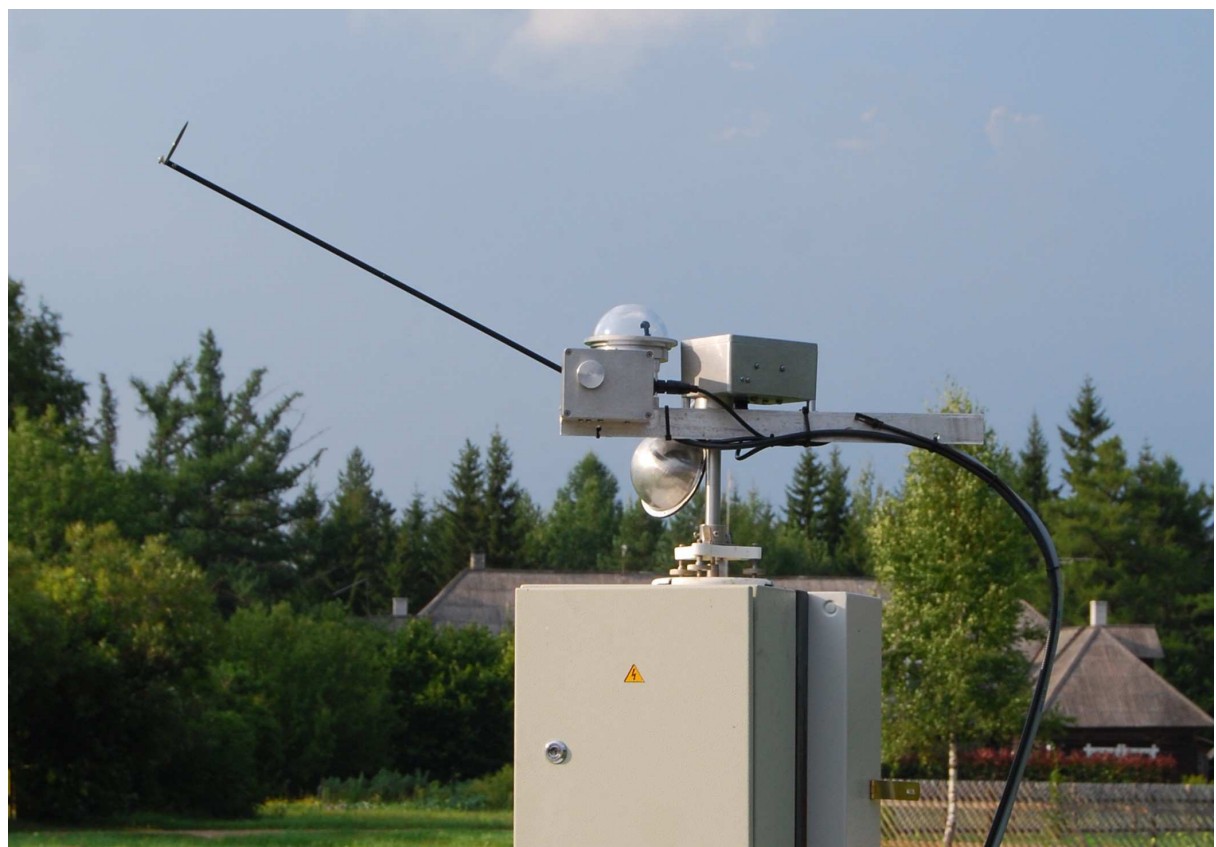

**Figure 1.** SkySpec at Järvselja, Estonia measuring diffuse sky irradiance.

**Table 1.** Summary of SkySpec measurements

| Year | Start date (dd.month) | End date (dd.month) | Number of measurement days |
|------|-----------------------|---------------------|----------------------------|
| 2013 | 08.08 | 22.11 | 102 |
| 2014 | 28.04 | 24.10 | 180 |
| 2015 | 05.05 | 24.11 | 219 |
| 2016 | 06.05 | 03.11 | 187 |
| 2017 | 12.04 | 21.11 | 229 |
| 2018 | 04.05 | 05.11 | 187 |
| 2019 | 15.05 | 22.11 | 192 |
| 2020 | 06.05 | 25.10 | 171 |

In total, measurements on 1467 days are considered in the analysis.



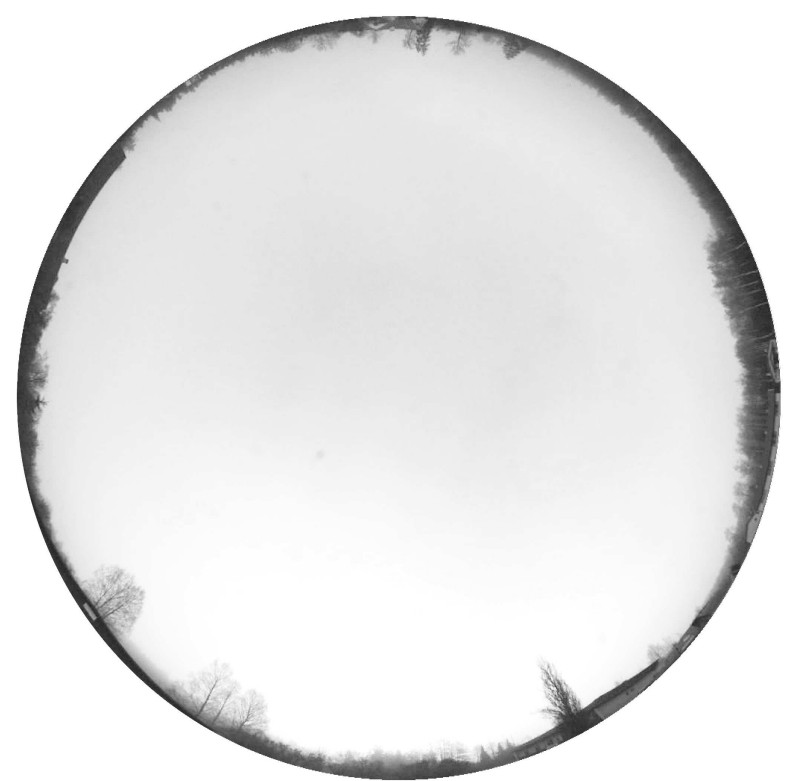

**Figure 2.** The cover of horizon at the spectrometer in 2013–2016.

291, 400, 526, 649, and 721, respectively, altogether 2587 spectra. The clear sky spectra were extracted manually, observing
that the global flux was stable during several minutes before and after the moment when the sun zenith angle was equal to one
from the prescribed set (40°, 45° etc).

## 4  Results

### 4.1  Global radiation

Mean spectra of global radiation in case of clear sky at various sun zenith angle are plotted in Fig. 3. Standard deviations are
marked by errorbars for sun zenith angles of 40° and 60°.

Kondratyev (1965) suggested an approximate solution of the radiative transfer equation for the calculation of spectral hemi-
spherical solar radiation during cloudless sky:

$$Q_\lambda(\theta_s) = \frac{S_0(\lambda)\,\mu_s}{1 + f(\lambda)\,/\,\mu_s}, \tag{1}$$



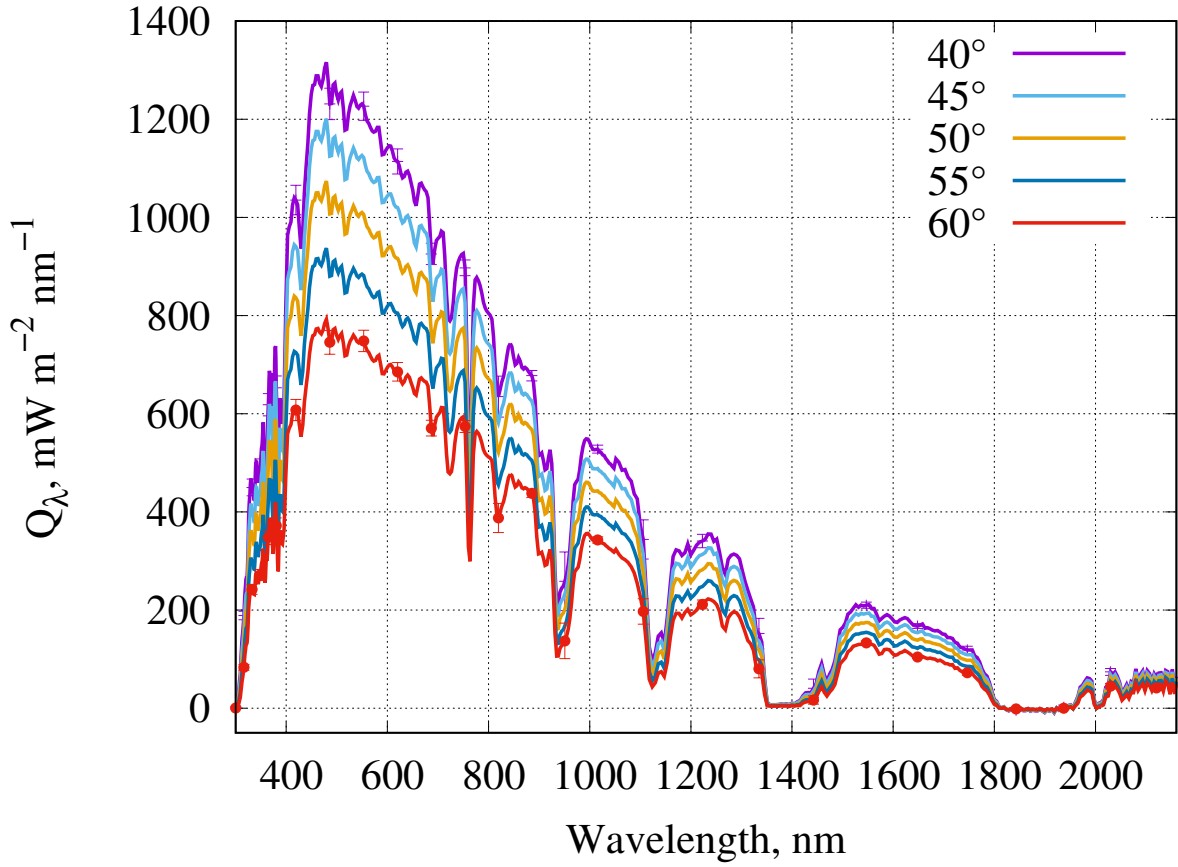

**Figure 3.** Mean spectra of global radiation in case of clear sky at various sun zenith angles.

where $\theta_s$ is the sun zenith angle, $S_0(\lambda)$ is the spectral solar constant, $\mu = \cos(\theta_s)$, and $f(\lambda)$ is a turbidity parameter of the
75 atmosphere. The approximation Eq. (1) is applicable at wavelengths of low absorption. The parameter $f(549.8)$ of the 2587 spectra varied between 0.0546 and 0.1864, while most of $f$-values are between 0.08 and 0.13. Smoothed distribution of the $f$-parameter is plotted in Fig. 4.

Mean spectra in Fig. 3 are rather similar but in differing scale. In Fig. 5 the ratio $Q_\lambda(\theta_s)/Q_\lambda(\theta_s = 40°)$ is plotted for different sun zenith angles $\theta_s$. Thin horizontal lines of same color mark the ratio according to the Eq. (1) for $f(\lambda) = 0.09$.
There are spectral intervals where this ratio is close to the theoretical value according to Eq. (1). Deviations of measured ratios from straight lines in Fig. 5 indicate the change in spectral composition of global radiation with changing sun zenith angle. Deviations are large in absorption bands. A systematic change is the decrease of the share of blue radiation with increasing sun zenith angle. Fig. 5 provides the quantitative measure for this change.

Principal component analysis of clear sky spectra in the spectral range 300–2160 nm for SZA from 40° to 60° revealed that
the first eigenvector describes more than 98.6% of spectral variability. Consequently the main change in the solar spectrum





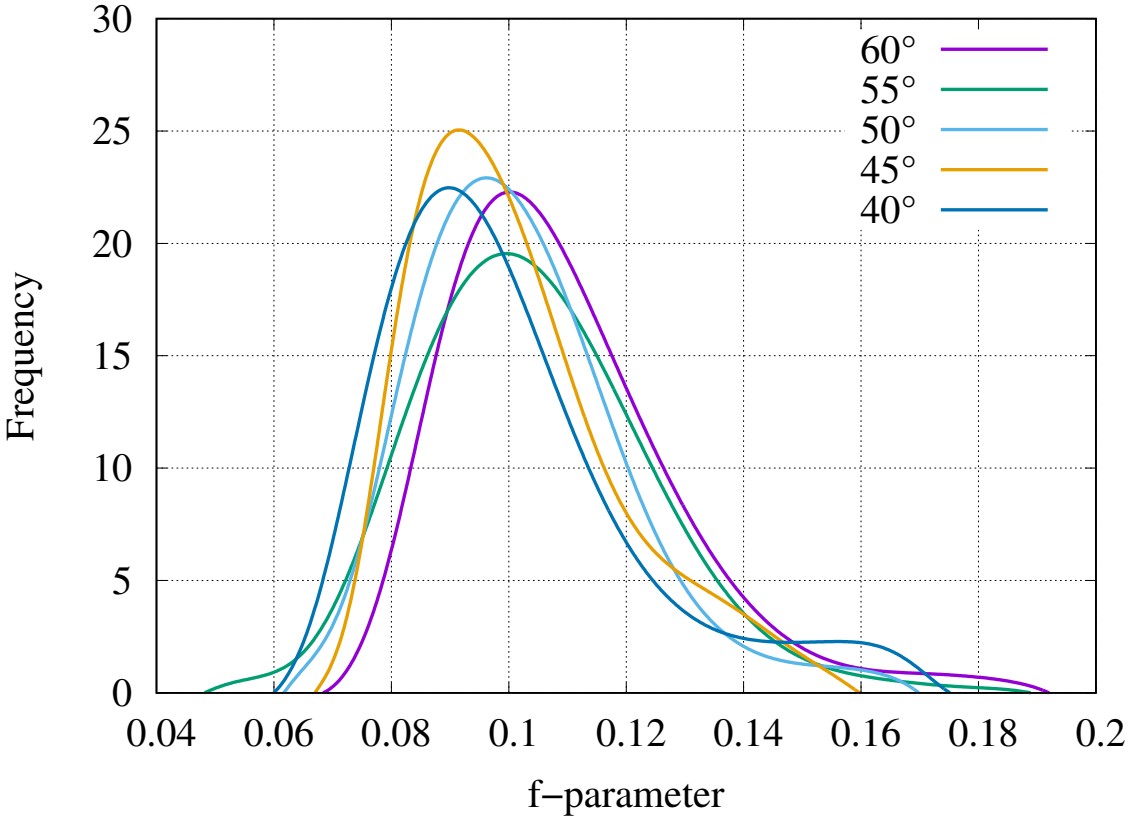

**Figure 4.** Distribution of $f(549.8)$ for various sun zenith angles.

is the change of flux level with changing turbidity, and changes in spectral composition are small. This conclusion and small deviations of measured curves from straight lines in Fig. 5 in rather wide spectral intervals allow to apply the approximation Eq. (1) for integrated solar flux in some rather wide spectral range as well. In Fig. 6 the distribution of relative global radiation $Q/Q_0$ for the whole measurement period (1467 days) in the wavelength range 320–1800 nm for years from 2014 to 2020 is plotted. The short measurement period of 2013 is not included in this plot. As the used spectral range includes absorption bands, the f-parameter value for the whole spectral range $f(\Delta\lambda) = 0.15$ was used for all years. In case of wrong $f$-value the peak in Fig. 6 moves away from the value $Q/Q_0 = 1$.

The main shape of the distribution of $Q/Q_0$ is formed by cloudiness. In general, cloudiness decreases the global solar flux compared to the flux in case of no clouds at the same sun zenith angle. Ratio values $Q/Q_0 > 1$ are caused by focusing of sun radiation by broken clouds, mainly by partial cover of Cu-clouds when direct sun radiation between clouds is not attenuated, and bright clouds near sun increase the diffuse sky flux substantially.

Variations of turbidity modify the peak of distribution at $Q/Q_0 = 1$. Small variations in the position of the peak may be caused both by different turbidity of the atmosphere in different years and by errors in the calibration of SkySpec. The summer





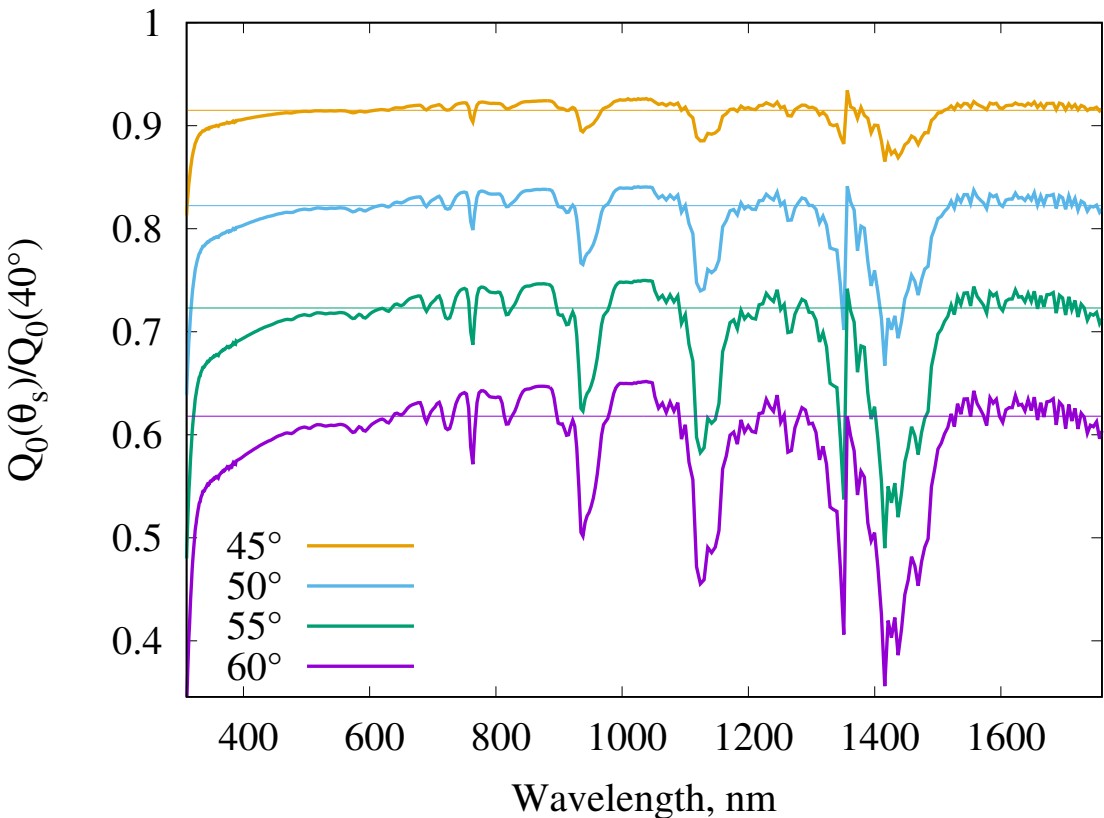

**Figure 5.** Ratio of mean spectra at various sun zenith angles to the mean spectrum at $\theta_s = 40°$.

of 2018 was very sunny, thus the peak at $Q/Q_0 = 1$ is higher and the level of the distribution at $Q/Q_0$ in the range between

0.1 and 0.5 lower than in other years.

Change of cloudiness modifies not only the relative global flux $Q/Q_0$ but also the distribution of energy in the spectrum of total radiation. With increasing cloudiness the share of spectral flux at some wavelengths increases but at some other wavelength decreases. Fig. 7 shows how change of cloudiness (change of relative global flux $Q/Q_0$) modifies relative share of radiation at various wavelengths in the integral flux.

**4.2    Diffuse radiation**

Energetic values of spectral diffuse sky radiation vary in wide range. Here we analyze the relative spectra of diffuse radiation as the ratio to the spectral global flux. The records of global and diffuse spectra are not simultaneous, therefore the spectral global flux at the moment of recording diffuse flux is calculated as the mean value of the last global spectrum before and the first one after the diffuse irradiance measurement stage.





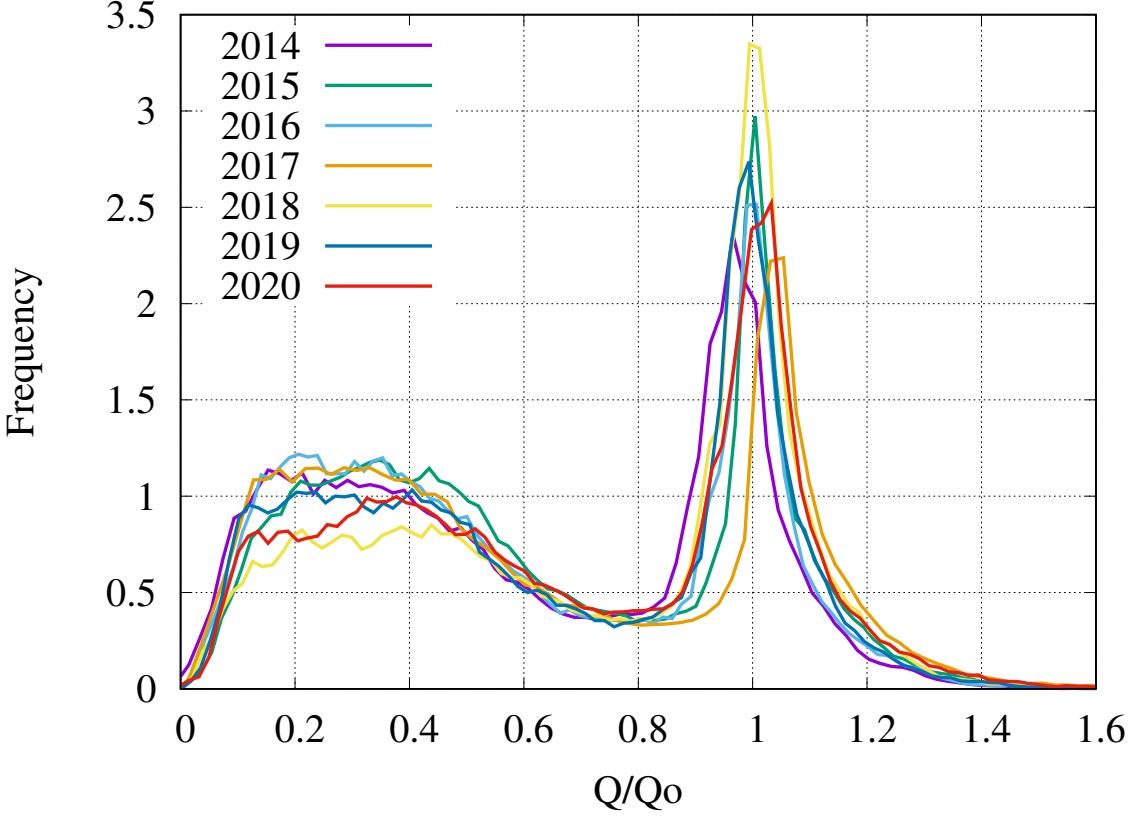

**Figure 6.** Distribution of $Q/Q_0$ in different years, $Q$ is the measured total radiation in the spectral range 320–1800 nm, $Q_0$ is calculated with Eq. (1) using $f = 0.15$.

Ratio of diffuse spectral flux to global flux during clear sky for various sun zenith angle is plotted in Fig. 8. Only data of
2015 are used in this figure. The selection of spectra was based on the ratio $Q/Q_0$, $0.95 < Q/Q_0 < 1.05$, see Fig. 6. In bands of water absorption at $\lambda = 1130$ nm and $\lambda = 1395$ nm the signal/noise ratio is low and the $D(\lambda)/Q(\lambda)$ ratio values in Fig. 8 are not reliable. Error bars mark standard deviation of spectra.

Figure 9 shows the $D/Q$ spectra at various $Q/Q_0$ values. Increase of cloud amount (decrease of $Q/Q_0$ ratio) increases the
general level of diffuse radiation, and also the balance of blue and NIR/SWIR radiation in diffuse flux. In case of $Q/Q_0 > 1$ the share of red and NIR-SWIR radiation increases substantially. Ratio values $Q/Q_0 > 1$ are caused by focusing of sun radiation by clouds. The exact ratio value depends on the cloud pattern near sun direction and dependent on weather conditions (wind speed at cloud level) may change rather fast. That will cause changes in the spectrum of diffuse sky flux as well.

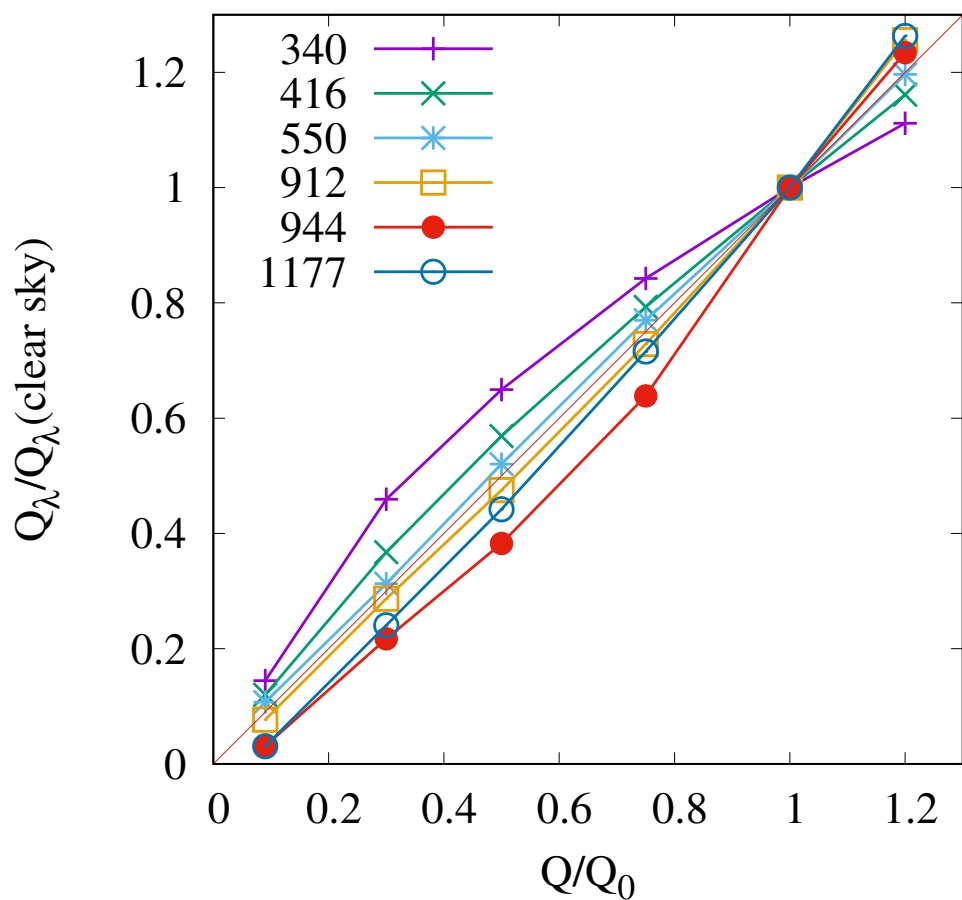

**Figure 7.** The change of share of spectral flux in the integrated flux at different wavelengths, sun zenith angle between $40°$ and $50°$.

## 5   Discussion and conclusions

Systematic spectral measurements of downwelling solar radiation, both global and diffuse, have been collected during 8 years in the hemi-boreal zone in south east Estonia near the SMEAR-Estonia research station. The measurements provided information about the variation of spectral and total fluxes of downwelling hemispherical global and diffuse solar radiation in the wavelength range from 300 to 2160 nm with spectral resolution of 3 nm in UV to 16 nm in SWIR spectral regions. Unique data have been collected and quantitative description of the variability of the measured spectra is provided. For the description of the synoptic situation during measurements the ratio $Q/Q_0$ in the spectral range of 320–1800 nm is used. We suggest to use this measure as the primary description of the weather situation instead of cloud cover. Cloud cover is estimated in meteorological stations





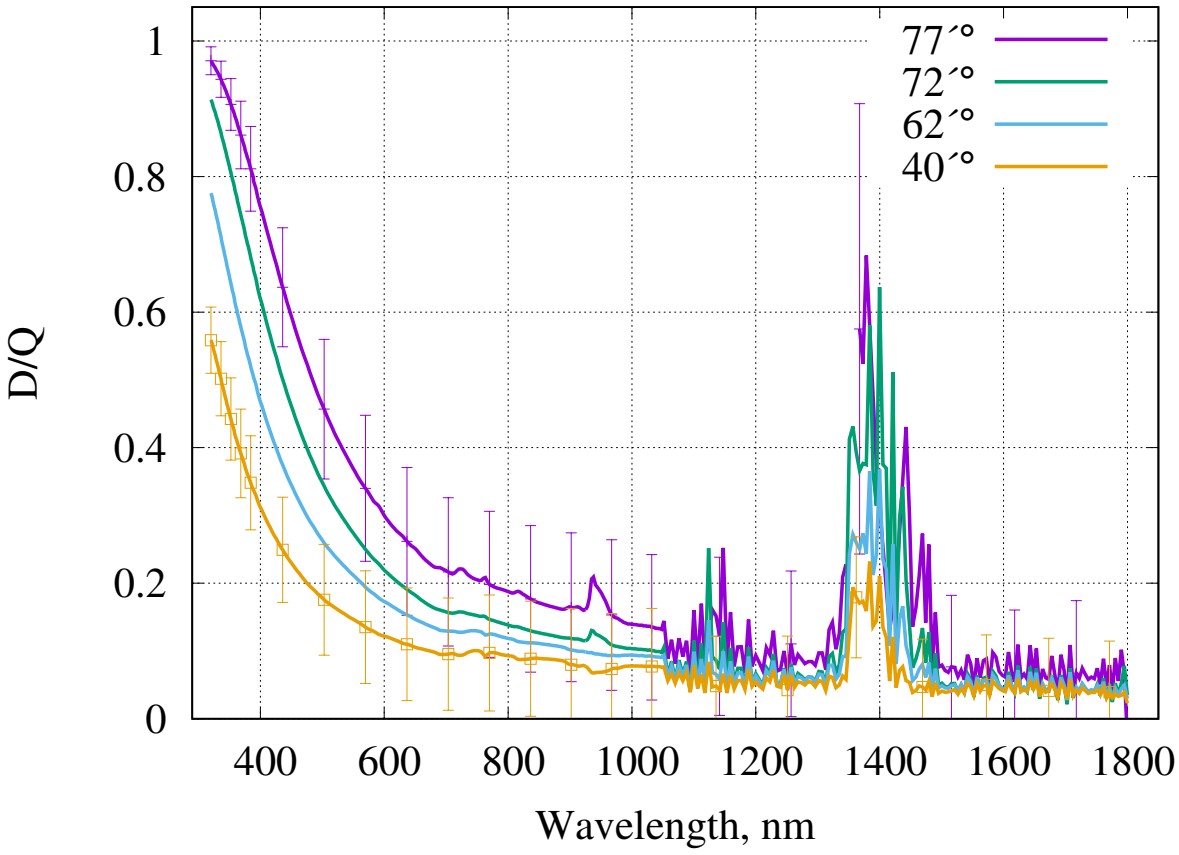

**Figure 8.** Ratio of diffuse to global spectral fluxes at various sun zenith angles during clear sky. Error bars mark standard deviation.

mainly visually, and thus may have subjective errors. At the same time the measurement of integrated global radiation with pyranometers is not very complex nor expensive. The number of meteostations where the measurements of downwelling solar radiation with pyranometers are carried out is increasing (Ohmura et al., 1998). Recording of downwelling solar flux is much

more simple than the instrumental measurement of cloud cover at a meteorological station.

*Data availability.* Spectra of downwelling global and diffuse solar radiation for the period and spectral range of interest are available at Tartu Observatory, Estonia. Please contact authors for arranging details.

*Competing interests.* The authors declare that they have no conflict of interest.





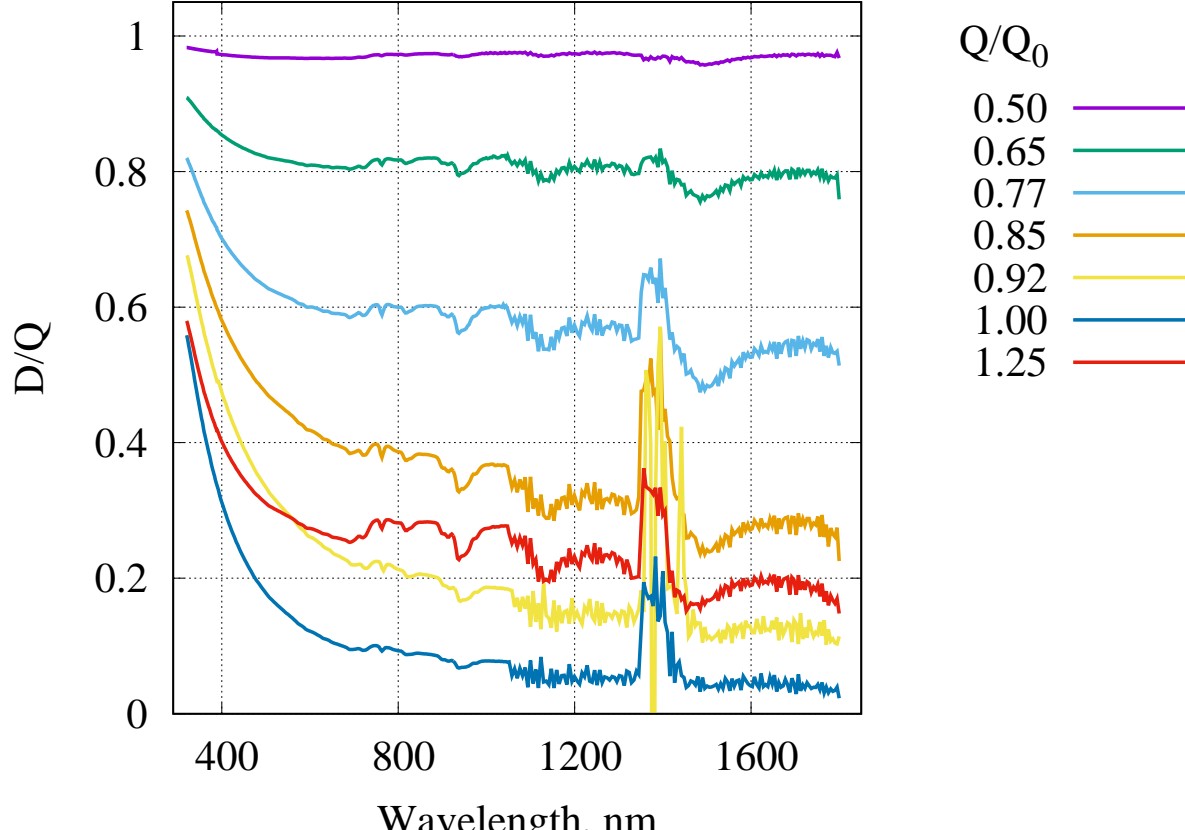

**Figure 9.** Ratio of diffuse to global spectral fluxes at various $Q/Q_0$ values, sun zenith angle $40°$.

*Acknowledgements.* This work was supported by Estonian Environmental Observatory, Project 3.2.0304.11-0395, by the EU Regional De-
velopment Fund, and by Estonian Research Council grants PUT 232 and PUT 1355.



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
