# Peer review of "Measurement report: Spectral actinometry at SMEAR-Estonia"

_Atmospheric Chemistry and Physics, 2020_

## Referee Comment (RC1) · Anonymous Referee #1 · 10 Feb 2021

Review of a paper entitled "Measurement report: Spectral actinometry at SMEAR-Estonia" by Kuusk and Kuusk

General comments:

The paper describes the results of 8 years of summer-season measurements of downward direct and diffuse spectral irradiances covering most of the solar spectral range at a station in Estonia. An analysis is presented for a limited range of solar zenith angles between 40 and 60 degrees for both clear-sky and all-sky conditions, mostly based on an approximate expression of atmospheric transmittance from the literature. While the dataset itself is certainly interesting and may be special as to its long-term character, the data analysis and conclusions drawn by the authors in my view are too sketchy and not adequately discussed, also in terms of previous studies. The apparent main

conclusion of the paper - that the ratio of measured to possible shortwave irradiance could be used as a meteorological parameter instead of cloud cover - is not justified. The work should not be published and the authors should reconsider their approach.

Specific comments:

Line 2: Specify that solar irradiances were measured. "Radiation" is no quantity.

Line 4: Specify that total irradiances were measured. Flux has a unit of power, irradiance of power per area. Check for the whole manuscript.

Line 8: "This ratio could be used as the primary meteorological parameter instead of cloud cover...". This suggestion will be discussed below where it is repeated (conclusions).

Lines 15-23: This paragraph of the Introduction lists some relatively recent studies showing that spectral composition influences plants which is not a new finding. The specific studies appear randomly chosen and have no direct association to the present work. Moreover, phrases like "Many trait allometries shift substantially due to plasticity across different growing conditions..." sound impressive but are misplaced in this work on radiation measurements.

Lines 23-25: The citation Moon et al., 2020 appears more appropriate in the context of a forest station if the authors think about using their measurement data as an input for a radiative transfer model within a canopy. This aspect could be discussed in more detail.

Line 29: In what sense Eddy-covariance measurements would benefit from spectral rather than broadband irradiance measurements remains unclear. I think the term "hyperspectral" is incorrectly used here. Hyperspectral measurements provide spectral and spatial information. In the present work downward irradiances were measured with and without direct sun contributions.

Line 42: The spectral ranges for which the resolutions are stated should be specified

more clearly.

Line 55: At least some basic experimental information should be given in a "measurement report": on the calibration source (traceable to?), the calibration frequency, the type of cos-receiver (e.g. integrating sphere), and the accuracy of the measurements. A comparison of integrated spectral irradiances with those recorded by the collocated pyranometers would also be reassuring.

Line 64: "The clear sky spectra were extracted manually, observing that the global flux was stable during several minutes ..." Certainly, stability was not the only criterion to characterize the situation as clear sky but also the irradiance itself. Conditions can be stable also under cloudy conditions. However, as long as the sun is not covered by a cloud, it is hard to say if the conditions were actually clear. This is an example where additional cloud-cover information would be helpful to classify the atmospheric state.

Line 74: The source of extra-terrestrial spectral irradiances that was used in the analysis should be referenced. Were annual variations considered and the instrumental spectral resolutions?

Lines 75-77, Fig. 4: What can be learned from the peak values and the variability of the  $f(\lambda)$  at 550 nm shown in Fig. 4? The authors introduce  $f(\lambda)$  as a "turbidity factor". However, the formula Eq. (1) is the solution for the two-stream approximation, assuming an asymmetry factor of 0 and  $f(\lambda)$  has a clear physical meaning: it is the atmospheric optical depth divided by two. Thus, a minimum value  $f(\lambda) = 0.048$  is expected for 550 nm for pure Rayleigh scattering (Bucholtz, 1995) for which the g=0 assumption is justified, i.e. in the absence of aerosols. Greater values as shown in Fig. 4 not surprisingly imply the presence of aerosols. However, aerosols are not mentioned in the whole manuscript. This is dissapointing because the measurement data of the authors allow to retrieve optical depths and aerosol optical depths widely independent of any approximations and assumptions.

Lines 80-83: The ratios of the spectra shown in Fig. 5 reveal nothing new to atmo-

СЗ

spheric physicists: stronger scattering towards smaller wavelengths and absorption features from ozone, oxygen and water vapor. The straight lines obviously fail to describe the wavelength dependent features.

Lines 84-92: It is surprising that based on the results shown in Fig. 5 the authors come to the conclusion that Eq. (1) is suitable to describe integrated clear sky spectral irradiances  $Q_0$  as well, using a somehow adjusted  $f(\Delta\lambda)=0.15$  which again only applies for the narrow range of solar zenith angles between 40 and 60 degrees (merely covering an airmass change of 0.7), an apparently relatively clean remote environment and comparatively low subarctic summer total water columns. The authors should note that the same analysis could have been done with pyranometer data. Moreover, this is not the first work dealing with parametrizations of global irradiance. No reference is made to other studies, no comparison is made with other approaches. See for example the recent review by Ruiz-Arias and Gueymard, 2018.

Lines 94-100: The strong influence of clouds on the global irradiance as shown in Fig. 6 is no surprise. The authors say that values above unity are caused by cloud reflections. This is a well-known phenomenon but it also shows that the ratio  $Q/Q_0$  alone is not a perfect indicator for the presence or absence of clouds. Similarly, there will be situations where the sun is blocked by single clouds and where the  $Q/Q_0$  indicate (local) cloudiness while in fact in a wider area (e.g. the forest) clear sky conditions prevail. By the way, values above 1 may also occur at low aerosol load and/or low water columns compared to the  $Q_0$  conditions.

Lines 101-104: This paragraph just states that the presence of clouds modifies the spectrum but not in which way. The data shown in Fig. 7 and the selection of wavelengths are not explained.

Lines 110-113: Why are the ratios in the range around 1400 nm consistently enhanced? I understand that S/N ratios are low in this region but that should affect the scatter and not the ratios. Or is there a physical explanation? See Fig. 9.

Line 114-118: What exactly is shown in Fig. 9? Are this single spectra or mean spectra binned over certain intervals of  $Q/Q_0$ ? Add standard deviations. The enhancement of the ratios around 1400 nm vanishes at low  $Q/Q_0$ . That should be explained.

Lines 120-130, Discussion and conclusions: This paragraph is virtually identical to the Abstract which is uncommon and certainly not acceptable in a scientific journal.

Lines 125-130: The main conclusion: "For the description of the synoptic situation during measurements the ratio  $Q/Q_0$  in the spectral range of 320–1800 nm is used. We suggest to use this measure as the primary description of the weather situation instead of cloud cover. Cloud cover is estimated in meteorological stations mainly visually, and thus may have subjective errors. At the same time the measurement of integrated global radiation with pyranometers is not very complex nor expensive. The number of meteostations where the measurements of downwelling solar radiation with pyranometers are carried out is increasing (Ohmura et al., 1998). Recording of downwelling solar flux is much more simple than the instrumental measurement of cloud cover at a meteorological station."

I disagree in several points. First, cloud cover was not measured in this work. So even if there were deficiencies of cloud cover measurements, the present work cannot reveal them. Second, cloud cover is a meteorological parameter that is not only relevant for solar radiation studies on the ground but also for other applications. Third, cloud cover is nowadays often recorded automatically with low-cost all-sky cameras and suitable software tools that even distinguish cloud types (see e.g. Lothon et al., 2019). Finally, also the data of this work show that cloud cover information would be helpful to better classify measurement conditions.

**References**

J. A. Ruiz-Arias and C. A. Gueymard, Worldwide inter-comparison of clear-sky solar radiation models: Consensus-based review of direct and global irradiance components simulated at the earth surface, Solar Energy, 168, 10-29, 2018.

A. Bucholtz, Rayleigh-scattering calculations for the terrestrial atmosphere, Applied Optics, 34, 2765-2773, 1995.

M. Lothon et al., ELIFAN, an algorithm for the estimation of cloud cover from sky imagers, Atmos. Meas. Tech., 12, 5519–5534, 2019.

---

## Referee Comment (RC2) · Anonymous Referee #2 · 10 Feb 2021

This study reports the 8-year summertime observations of downwelling solar radiation in a research station in southeast Estonia. The observations include radiation across a range of spectrum and of both global and diffuse radiation. Some basic examinations and analyses of the observations from different perspectives are conducted. Although the data and the report themselves are useful, the manuscript lacks scientific insights. The major conclusion is that the ratio of the measured flux to possible total flux can dictate the cloud cover. This result itself is of little interest to the community, not to mention that it is subject to very weak justification (lack of actual cloud cover measurements) and is not fully discussed in terms of its potential impacts on the field. Although I understand that this manuscript, as a measurement report, is not science-oriented, the level of the scientific quality does not meet the basic standard of ACP for any kind

of publication. For these reasons, I suggest rejection.

---

## Author Comment (AC1) · 24 Mar 2021

Dear Prof. Zhanqing Li,

Handling Editor

We thank Atmospheric Chemistry and Physics for keeping our preprint on-line.

We have received two proposals for collaboration:

On 22 Dec 2020: "I find your measurements of the highest value because of the extended spectral range and the sensing of both the global and diffuse spectral distributions, which is extremely rare. [..] I'd like to discuss with you about being granted access to your data, ..."

On 10 Feb 2021: "... our studies have been limited to the 350-1050 nm waveband due to the limited range of measurement of our spectroradiometers (EKO MS-700). In this sense, we would be interested in exploring if APE and/or some other spectral indexes (e.g.: Blue Fraction) might verify a one-to-one relationship with the spectrum shape using some other 'extended spectrum' datasets (up to 1800-2000 nm). [..] we would be happy if you accepted to collaborate with us in our studies by granting access to your data and contributing with your knowledge."

Two invited referees suggest to reject the submission:

RC1: The work should not be published

RC2: I suggest rejection.

Thus, we do not expect that ACP will publish our measurement report,

and we terminate sharing our data via ACP.

Sincerely yours,

Andres Kuusk

Tartu Observatory, Estonia

andres@to.ee